# A Contemporary Review on Utilizing Semantic Web Technologies in Healthcare, Virtual Communities, and Ontology-Based Information Processing Systems

**Senthil Kumar Narayanasamy** [1], **Kathiravan Srinivasan** [2], **Yuh-Chung Hu** [3], **Satish Kumar Masilamani** [4] **and Kuo-Yi Huang** [5,*]

1   School of Information Technology and Engineering, Vellore Institute of Technology, Vellore 632014, India; senthilkumar.n@vit.ac.in
2   School of Computer Science and Engineering, Vellore Institute of Technology, Vellore 632014, India; kathiravan.srinivasan@vit.ac.in
3   Department of Mechanical and Electromechanical Engineering, National ILan University, Yilan 26047, Taiwan; ychu@niu.edu.tw
4   Sri Venkateshwara College of Engineering & Technology, Chittoor 517127, India; m.satishkumar@svcet.in
5   Department of Bio-Industrial Mechatronic Engineering, National Chung Hsing University, Taichung 402, Taiwan
*   Correspondence: kuoyi@dragon.nchu.edu.tw

**Abstract:** The semantic web is an emerging technology that helps to connect different users to create their content and also facilitates the way of representing information in a manner that can be made understandable for computers. As the world is heading towards the fourth industrial revolution, the implicit utilization of artificial-intelligence-enabled semantic web technologies paves the way for many real-time application developments. The fundamental building blocks for the overwhelming utilization of semantic web technologies are ontologies, and it allows sharing as well as reusing the concepts in a standardized way so that the data gathered from heterogeneous sources receive a common nomenclature, and it paves the way for disambiguating the duplicates very easily. In this context, the right utilization of ontology capabilities would further strengthen its presence in many web-based applications such as e-learning, virtual communities, social media sites, healthcare, agriculture, etc. In this paper, we have given the comprehensive review of using the semantic web in the domain of healthcare, some virtual communities, and other information retrieval projects. As the role of semantic web is becoming pervasive in many domains, the demand for the semantic web in healthcare, virtual communities, and information retrieval has been gaining huge momentum in recent years. To obtain the correct sense of the meaning of the words or terms given in the textual content, it is deemed necessary to apply the right ontology to fix the ambiguity and shun any deviations that persist on the concepts. In this review paper, we have highlighted all the necessary information for a good understanding of the semantic web and its ontological frameworks.

**Keywords:** semantic web; ontology; Web 2.0; information retrieval; virtual communities; e-learning; healthcare

## 1. Introduction

While discussing the potential of the web, particularly Web 2.0 or the recently evolving semantic web, the foremost thing to remember is the network effect. This phenomenon was witnessed in the most popular PageRank algorithms [1] developed by Google to assign priority to web pages and enhanced the searching and retrieval operations on its search engine. The implementation of the PageRank algorithm has solved the impending problems of keyword-based search; as a result, retrieval performance has been increased gradually through appropriate indexing schemes. In this paper, we highlight the difference that exists between Web 2.0 and the semantic web in terms of linked spaces created. Primarily,

Web 2.0 [2] allows connecting the web users but not their contents. On the contrary, the semantic web [3] creates a semantic space to link web users as well as their posted content online. The semantic web takes a significant advantage in creating links between different ontological sources and designing the joint network effects to connect the social space for its web users. The mutual coupling of Web 2.0 and the semantic web boasts the new trend for web users to explore the social networks and enables them to create a semantic space for annotated content. The Web 2.0 content has become dynamic through technologies such as AJAX, Web Services, and other inherent tools. Before the advancement of Web 2.0, websites such as Flickr and Del.icio.us permitted web users to create their content dynamically and post on these sites. This basic idea of creating content by web users is paving the way for the emerging Web 2.0. Earlier, blogs, dashboards, forums, and Wikipedia attained considerable success in this aspect and made use of RSS, permalinks, and some implicit technologies to make this factor achievable. The author O'Reilly [4] highlighted that the link space that emerged from the blogs and other sites enabled the network effect and made the web content more dynamic.

Likewise, when discussing the prevalent utilization of Web 2.0 components, the technological revolutions have faced a tremendous focus in recent days rather than the social phenomena over Web 2.0 applications. Over the years, social networking sites such as Facebook, LinkedIn, Twitter, Instagram, and MySpace have achieved phenomenal success and received massive attention among the general audience. The success of these social sites has relied mainly on content sharing, and features of content sharing can be achieved through social connections instead of web search or query optimization methods [5]. In the last few years, YouTube has been performing exceedingly well and let social users upload their videos onto the web for free. The most exciting features added to YouTube are the inclusion of email and blogging facility, which enables social users to share videos with their friends or relatives and make their video a popular one in the network of videos. Here, the popularity of the video has been enhanced not by the content of the video but instead based on the sharing features as well as the metadata features of the videos. The searching on YouTube has been made through the social connection but not by the semantic content of the videos, such as what has been discussed in the video. Moreover, primary indexing is created for all the videos, and the social overlay of the videos is carried through the primary indexing mechanism. The list of abbreviations used in this manuscript along with their full form is given in Appendix A.

### 1.1. Contribution of This Survey

In this survey paper, we have delineated the existing approaches followed for the use of semantic web technologies and highlighted the applications that have been pervasively deployed using semantic web models. The key contributions of this review are summarized as follows:

- We provide a contemporary review on utilizing semantic web technologies in healthcare, virtual communities, and ontology-based information processing systems;
- We discuss the lack of reusability of the applications such as e-learning and healthcare datasets from a semantic web technologies perspective;
- We elaborate on the poor resource-sharing mechanisms between application-specific models, particularly in virtual communities and information retrieval;
- We discuss the absence of real-time availability of data in semantic web technologies;
- We present various open challenges and numerous future research directions for semantic web technologies.

In this connection, detailed work has been carried out and has identified the pitfalls observed by several researchers in these fields, as compared meticulously in Table 1.

**Table 1.** Comparison with previous surveys related to semantic web technologies (✔: available, ✕: not available).

| Authors, Reference and Published Year | Count of Articles | Time Duration | Systematic/ Scoping Review | One Phrase Summary | Discussion on Real-Time Availability of Data | Elucidation on Open Issues/ Challenges | Explication on Future Directions/ Road Ahead |
|---|---|---|---|---|---|---|---|
| Our Review | 65 | 2011–2021 | Systematic | This review presents the domain-specific advances reached in semantic web technologies and offers comprehensive methodologies followed in healthcare, virtual communities, and ontology-based information processing systems. | ✔ | ✔ | ✔ |
| Pascal, H. [6] 2021 | 39 | 2001–2020 | Scoping | This paper provides deep insights related to offering security for web services and is majorly focused on utilizing the OWL frameworks. | ✕ | ✕ | ✔ |
| Patel and Sarika Jain [7] 2021 | 25 | 2004–2018 | Scoping | The authors give the novel proposal of using named graphs to connect the potential entities and infer the contexts based on text co-references. | ✕ | ✕ | ✔ |
| Kurteva et al. [8] 2021 | ✕ | ✕ | Systematic | The contextual references for text documents can be inferred through the semantic space; this paper focuses more on following the best practices of using the semantic web tools. | ✕ | ✕ | ✔ |
| Yahya et al. [9] 2021 | 51 | 2010–2020 | Systematic | As this decade has been dominated by Industry 4.0, the dominance of using semantic web technologies is growing exponentially; this article gives the closer picture of creating the knowledge base such as DBPedia, FreeBase, and YOGO. | ✕ | ✔ | ✕ |
| Rhayem et al. [10] 2020 | 37 | 2011–2020 | Systematic | The intrusion of semantic web frameworks has even reached into the Internet of Things as well. This study provides the details pertaining to making meaningful services for IoT-enabled devices. | ✕ | ✔ | ✕ |
| Drury et al. [11] 2019 | 74 | 2002–2018 | Systematic | In recent years, precision agriculture has gained huge research attention and the occupation of the semantic web has even made it resilient and yields affordable solutions to many of the impeding problems faced in agricultural sectors. | ✕ | ✕ | ✔ |
| Moussallem et al. [12] 2018 | 53 | 2001–2017 | Systematic | Making machines to comprehend the human language has been a decadal issue in the research community. However, after the advent of semantic web technologies, it has become easier for machines to understand the text or image content. This paper gives a novel approach to converting text documents into predefined machine-translated structures. | ✕ | ✔ | ✕ |

*1.2. Survey Methodology*

This survey paper was constructed based on three important stages: (i) determination of the right scope of this review; (ii) extraction of research papers related to the semantic web, information retrieval, E-learning, and healthcare; (iii) report of the review based on some empirical evidence. In the first stage of this review process, the scope was determined based on the work carried out on this field between the year January 2011 and October 2021. In this connection, the research papers were selected based on the potential semantic-web-based keywords, which were searched for in different databases such as ACM Digital Library, IEEE Explore, Science Direct, SCOPUS, Google Scholar, etc. In the second stage, the advanced filtration process was carried out to rightly adapt to the titles and abstracts present in those research papers. In conjunction to this, papers related to some selected ontologies such as FOAF, SKOS, and Dublin-Core were identified for further review process. The third stage involved consolidating the research papers that possess strong empirical evidence and used some standard datasets for their implementation work. The full-text-reading approach has been strenuously carried out and finally selected only 65 articles to report this survey work. Additionally, we applied the Preferred Reporting Items for Systematic Reviews and Meta-Analysis (PRISMA) model to give the categorical depictions of the papers used in this survey.

### 1.2.1. Search Strategy and Literature Sources

For this extensive review, we searched for the keywords in the databases including Google Search and Web of Science journals, starting from January 2011 to October 2021. The potential search queries used in this survey study were "Semantic Web, Ontology, Web 2.0, Information Retrieval, Virtual Communities, E-Learning, healthcare". The research papers pertaining to the above keywords were analyzed and scrutinized for this survey paper. Specifically, the pragmatic approach of dealing with semantic web technologies in recent applications such as information retrieval and healthcare domains has been a focus of this paper. Likewise, the deep inclusion of semantic web frameworks in e-learning platforms and some virtual communities has received huge attention due to its overwhelming progress after the impact of COVID-19 pandemic.

### 1.2.2. Inclusion Criteria

We surveyed the research articles published between January 2011 and October 2021 in the field of semantic web technologies with domain applications such as healthcare, virtual communities, and ontology-based information processing. Moreover, we only considered research articles written in English and provided the empirical analysis on the role of semantic web technologies in the above-stated domain.

### 1.2.3. Elimination Criteria

Articles that have not been written in English, those reported before January 2011, case reports/case series, letters to the editor, opinions, commentaries, conference abstracts, dissertations, and theses were excluded from this review.

### 1.2.4. Results

To provide a comprehensive understanding of the evolving semantic web technologies in the fields of information retrieval, e-learning platforms, healthcare, and other virtual communities, we gathered almost 683 research articles and reviewed the abstracts and empirical frameworks used in those papers. This study includes peer-reviewed journals, conferences, and book chapters. Later, we removed 221 papers after title/abstract screening and considered only 462 research articles. After reviewing those articles thoroughly, we removed 243 articles that possess similar datasets, the same ontological networks, and/or similar implementation methods. Furthermore, we excluded 154 articles after full-text scrutiny. Eventually, we considered only 65 research articles that have been used with some standard techniques with good governance of most standard datasets released by

authorised agencies. The PRISMA flow diagram has been given in Figure 1 to highlight the seminal searching tasks rendered for this survey paper.

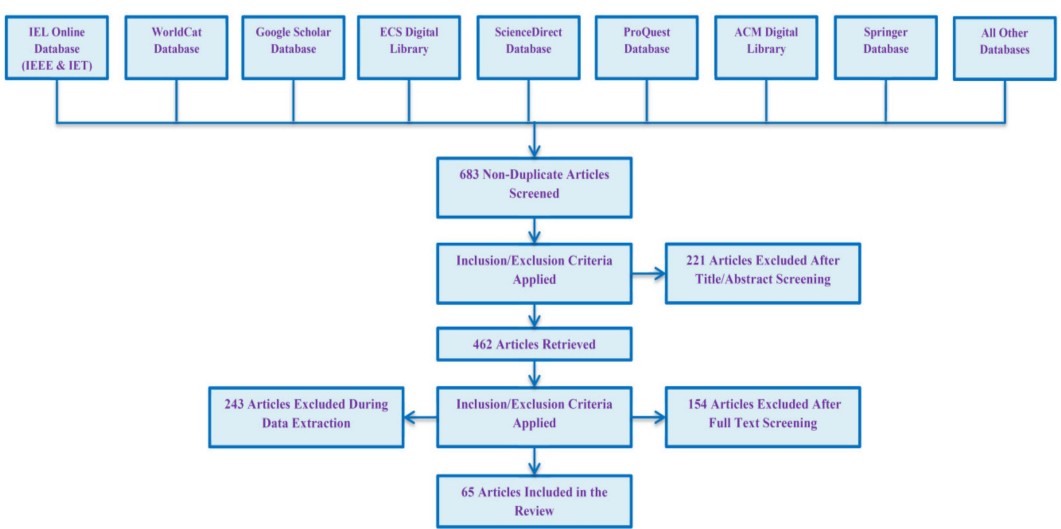

**Figure 1.** PRISMA flow diagram for the selection process of the research articles used in this review.

In Figure 2, the actual distribution of research articles related to the semantic web has been projected between the years January 2011 and October 2021.

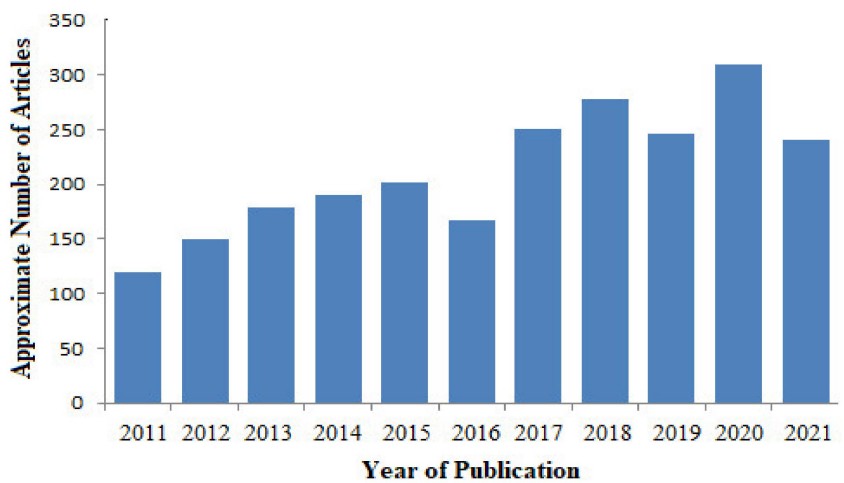

**Figure 2.** Year-wise distribution of retrieved papers for semantic web technologies in healthcare, virtual communities, and ontology-based information processing.

*1.3. Structure of This Survey*

The structure of this survey paper is organized as follows: Section 1 discusses the introduction of Web 3.0 and other semantic web technologies used in different applications. We briefly explain the survey methodology followed for this research article. In Section 2, we discuss the emergence of semantic web technologies and the right utilization of metadata to bring the ontological framework for the applications. Section 3 delineates the role of the semantic web in virtual communities and how it establishes the mutual connection over E-learning platforms. As E-learning platforms have been gaining huge attention post-COVID-19 pandemic, the deployment of ontologies into this e-learning network would be a great benefit for many real-time application developers. Section 4 highlights the role of ontologies in information retrieval tasks and explains the process of implementing ontologies for user-query refinement and information-linking approaches. Section 5 describes the impact of

semantic web technologies on healthcare datasets and binds the strong connection between different stakeholders such as doctors, patients, systems, and hospitals. Finally, Sections 6 and 7 give an overview of some of the open challenges faced in implementing semantic web technologies and the road ahead for the developers to think beyond this limit. The complete structure of this survey paper has been given in Figure 3.

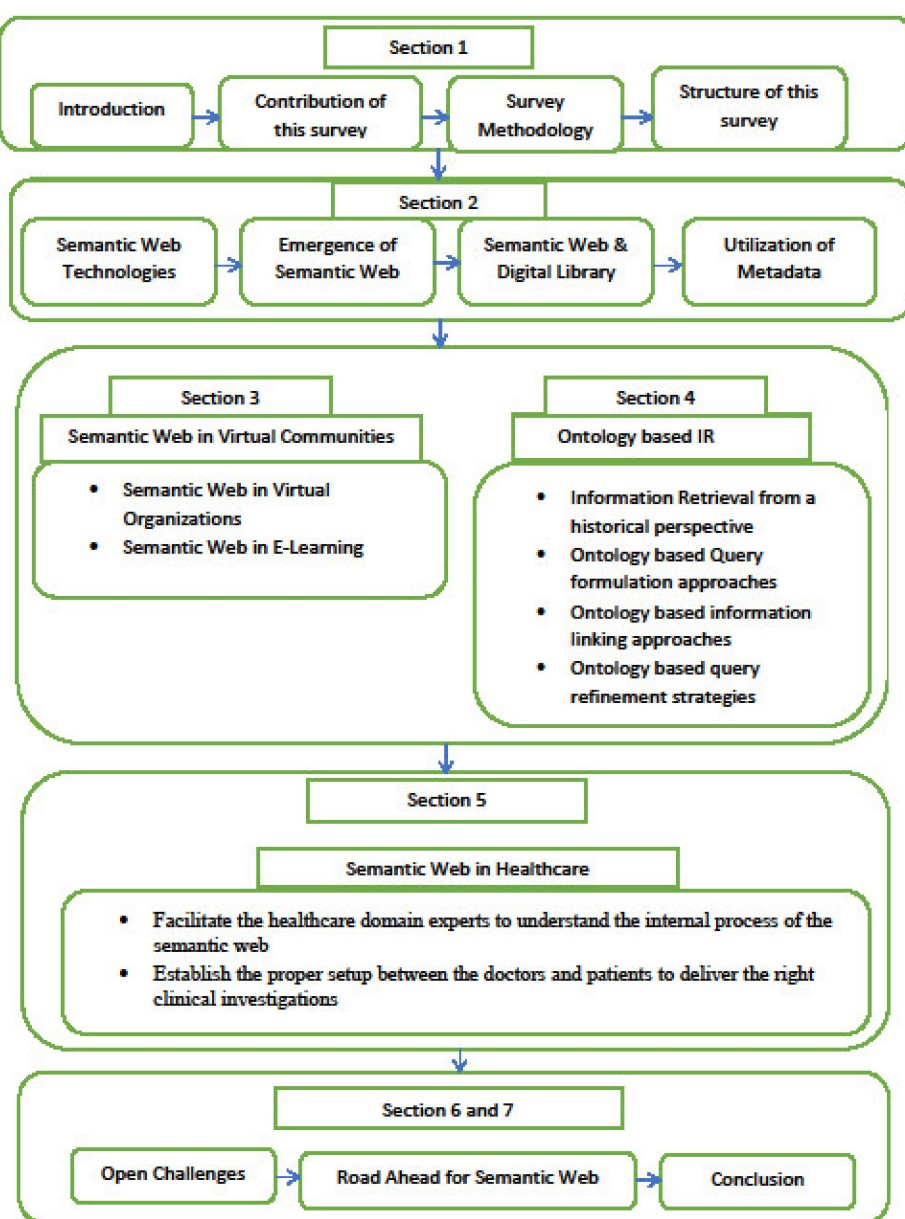

**Figure 3.** Block diagram representing the flow of this review.

## 2. Semantic Web Technologies

The semantic web has entirely redefined the process of converting web content into a more structured format so that the user's web query can be achieved with more accuracy and providing the intelligent system to integrate the data from diverse sources. Fundamentally, the strength of the semantic web majorly relies on its effective ontological connections to appropriately represent the information that is suitable for a machine-readable format. The ontologies are the data models used to connect the concepts through its potential named entities (i.e., classes and relationships). Normally, the classes are recognized as entities and relationships are the properties between two classes. Our aim is to bring the major issues pertaining to semantic web technologies and particularly the vast opportunities laid

forward for the correct utilization of web resources. In this connection, this survey provides a comprehensive overview of various datasets used for creating the knowledge sources and highlighting the major benefits of using these ontologies and datasets. Table 2 gives the various datasets used for semantic-web-related projects. Interventionary studies involving animals or humans, and other studies that require ethical approval, must list the authority that provided approval and the corresponding ethical approval code.

**Table 2.** List of various semantic web datasets.

| Reference Number | Year | Authors | Dataset Used | Total Documents | Topic |
| --- | --- | --- | --- | --- | --- |
| [13] | 2009 | Kulkarni et al. | IITB | 103 | Hybrid |
| [14] | 2011 | Hoffart et al. | AIDA/CoNLL | 57 | News |
| [15] | 2011 | Ritter et al. | Ritter | 2394 | News/Tweets |
| [16] | 2013 | Cano Basave et al. | Microposts2013 | 4265 | Tweets |
| [17] | 2014 | Carmel et al. | ERD2014 | 91 | Web Queries |
| [18] | 2014 | Cano et al. | Microposts2014 | 3395 | Tweets |
| [19] | 2015 | Derczynski et al. | Derczynski IPM NEL | 182 | Tweets |
| [20] | 2016 | Rizzo et al. | Microposts2016 | 9289 | Tweets |
| [21] | 2016 | Nuzzolese et al. | OKE 2016 Task 1 | 253 | Hybrid |
| [22] | 2018 | Szarnyas et al. | SNB | 250 | News/Tweets |

*2.1. Emergence of Semantic Web*

Some of the original motivations for the semantic web came from the early web applications that cause the problems for search and browsing in Web 2.0 applications. Latent semantics [23], an attempt to "mine" meaning from the words in web content, is always problematic due to its wide ambiguity and prevalent polysemy (the many meanings of a single word such as "run" or "left"). The class and subclass relations, which are crucial to language use, are also problematic. The semantic web technologies were created to deliver solutions for the faults that happened in Web 2.0. The significant finding that the semantic web has developed in recent years is that the applications are deemed to share much-sought information. Still, if that information is not in the textual format or is in a format that means that extracting the potential facts is difficult, then a suitable knowledge extraction pattern is required in the form of semantic web technologies [24]. However, these findings are not new to emerging fields such as natural language processing and machine translation [25]. Over the years, NLP and MT worked on these findings and fixed problems such as ambiguity of text, missing fields, part-of-speech (POS tagger) confusions, and many more. However, the utilization of semantic web technologies has helped to uncover seminal knowledge representation in the text that was deeply ingrained in the forms of entities and relationships. Moreover, this paves the way for very efficiently connecting the entities with appropriate web resources, as depicted in Figure 4.

This phenomenon has been achieved with a web graph: a graph exists between potential entities extracted from the textual sources, and the real-world entity persists on the web sources. To serve this purpose, semantic web languages such as resource description framework (RDF)/resource description framework schema (RDFS) and web ontology language (OWL) were used, and to shun the ambiguity persisting in the textual content, the semantic web languages mostly denote the terms or entities with assigned uniform resource identifiers (URIs) in the web, as given in Table 3. While much is said about the official capacities of these dialects and their ability to communicate individual connections, a substantially more basic perspective is that they can be utilized to give ordinary referents. Among the semantic web vocabularies [26], the friend-of-a-friend (FOAF) ontology has been widely used in textual processing to obtain the correct user references and link the entities with each other through the appropriate standard vocabularies. While inference is a significant part of the web and all different information portrayal dialects, the capacity

for connected terms is a fundamental contrast between RDF-based dialects and prior KR dialects.

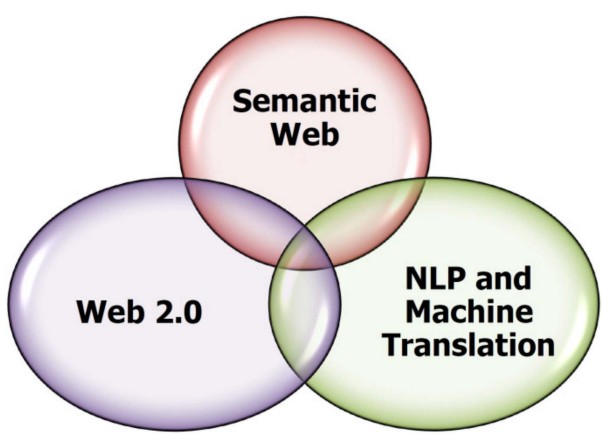

**Figure 4.** Connecting semantic web into Web 2.0, NLP, and machine translation.

**Table 3.** Commonly reused ontologies for web content.

| References | Core Ontologies | Ontology Description | Benefits | URL |
|---|---|---|---|---|
| [27–29] | RDF | Represent the potential real world named entities in the text | Used to identify the potential named entities and links into the semantic graph for easy traversal and disambiguation. | http://www.w3.org/1999/02/22-rdf-syntax-ns# (accessed on 22 December 2021) |
| [24,26,30] | Dublin Core | Set of metadata items that identifies the wide range of Web resources | Able to find the suitable metadata elements and assign the appropriate attributes for the values to distinguish the disparity. | http://purl.org/dc/terms/ (accessed on 22 December 2021) |
| [31–33] | FOAF | Associates people and relationship together and coupled them for interlinking | Very useful to segregate only PERSON entities from the document and categories the relationship between one or more entities. | http://xmlns.com/foaf/spec/ (accessed on 22 December 2021) |
| [34–36] | SIOC | Online communities for linking | Able to link the social communities into the real-world mentions and helps to classify the data based on the tags associated with the element. | http://semanticscience.org/resource/ (accessed on 22 December 2021) |
| [29,37,38] | SKOS | Represent the knowledge for the organization and any systems | Capable of filtering the hidden patterns and knowledge present in the text corpus. | http://www.w3.org/2004/02/skos/core# (accessed on 22 December 2021) |

The potential entities extracted from the documents using semantic web technologies have been very well interlinked with one another in many ways. Using the appropriate ontology, the entities were directly linked to the corresponding web URIs and reduced the ambiguities that persist on the multiple real-world entities. For instance, Flickr uses semantic web technologies to incorporate the site entities with appropriately labeled mentions in the OWL ontology [39]. The proper linking between two or more entities can be assigned with their own URIs and links those URIs in the Flickr site for easy access to resources and materials. This process can be easily achieved in ontology because of its basic vocabularies such as FOAF, Dublin Core, SIOC, etc. [40]. Furthermore, it creates a graph space to connect any entity with other entities in the network with ease. The graph space [41] grows with the proper utilization of RDF and OWL codes, and the emergence of more terms in the

documents will lead to the mapping of more links to be created and extracted from the knowledge base such as DBPedia and YAGO.

As the link space in the networked graph grows exponentially, the exploitation of links in the social construct is also becoming highly utilized. The semantic web tools [27] fail to focus on these interlinking mechanisms and provide fewer inference capabilities as long as the collection of entities is stored into a single triple store. Recently, new semantic web tools have been developed to incorporate these difficulties and make the graph space more explicit to provide the appropriate links to the entities that are stored in a single triple store. In particular, tools such as Tabulator and Zitgust [34] were implemented in the browser itself and made the graph space very sparse to accommodate the changes in the links. Specific applications help to create the potential relationships between the entities and then beginning to change this by giving programs that follow these connections, making the chart space progressively unequivocal. Another issue for the semantic web is that, up until this point, applications have not, to a great extent, abused the social components that control the Web 2.0 destinations. Very regularly, semantic web scientists have been centered on attempting to, some way or another, use labeling, and folksonomies in their present level and uncertain structure and have overlooked what is essential; this is the space where semantics is required and can most effectively be abused. Alternately, rather than abusing the network settings, intrigue gatherings and individual connections that make locales, for example, Flickr work or the mind-boggling social elements of Wikipedia, numerous semantic web applications center exclusively around master frameworks such as applications with expressive semantics to the avoidance of all else. These frameworks utilize the way that OWL obtains a norm, and in this manner, offer points of interest in that regard. However, they are not misusing the web idea of the semantic web.

A particular significant case to the above is the friend-of-a-friend ontology [31], which is without question one of the accomplishments of the semantic web to date. FOAF initially evolved as a little philosophy to depict individuals and to let them connect in an informal-organization-like way. FOAF was intended to be moderately lightweight and straightforward to utilize instead of pushing for a strong representation of people's properties. Specifically, utilizing RDF's idiom "seeAlso" was created to permit FOAF entities to connect and make an informal organization. Most FOAF documents are currently made naturally by other web destinations, for example, social media sites, and in this way, the quantity of these entities (and in this way, the estimation of the associations between them) develops quickly [1,23]. There are a huge number of FOAF profiles which, when contemplated over, include associations among the informal communities created from various sites. FOAF is to a great extent more productive due to its demonstration of the informal organizations that it encodes, although the connection space is still not as extensive as some Web 3.0 sites, and there is still a ton of exertion going into figuring out how to make all the more connections of FOAF to different ontologies, and more cases, to build the worth the system impact brings.

Notwithstanding the capability of connecting phrasings between entities, there is an element of sharing that is being made conceivable by the semantic web. As of now, there are various tasks centered on making high-worth datasets accessible in RDF to make them progressively accessible for applications to misuse. The basic semantics of these RDFized datasets make them simple to connect to and to depict utilizing the more expressive development of RDFS, OWL, and the rising standard dialects. For instance, the BBC has discharged its program inventory in an RDF good structure [28]. This makes 75 years of BBC programming accessible for connecting to semantic web sites. Consequently, for instance, it would be simple for RealTravel to connect to all the BBC shows occurring in, or providing details regarding, the known areas. This would facilitate the connection to Dopplr, Flickr, Wikipedia, MySpace, etc. The potential system impact made by connecting the URI space of web assets, the informal organizations of current Web 2.0 applications, and the URIs in these vocabularies is gigantic: Metcalfe's law [42], misusing the potential linkages of substance in these numerous spaces, predicts a staggering worth.

### 2.2. Semantic Web and Digital Libraries

While implementing the semantic web projects, digital libraries are the key segment of the data foundation which supports higher education and research activities largely. A key perspective for the digital library [35] is the arrangement of shared indexes that can be distributed and pervasively examined. This requires the utilization of basic metadata to depict the fields of the inventory (for example, creator, title, date, distributor) and controlled vocabularies to permit subject identifiers to be attributed to publications. By distributing controlled vocabularies in a single spot, which would then be able to be linked to all clients over the web. The library indexes can utilize similar web-available vocabularies for listing and increasing things with the most applicable terms for the space of concept hierarchy. At that point, web search tools [43] can utilize similar vocabularies in their pursuit to guarantee that the most applicable data details are returned. Figure 5 illustrates the concept of digital libraries.

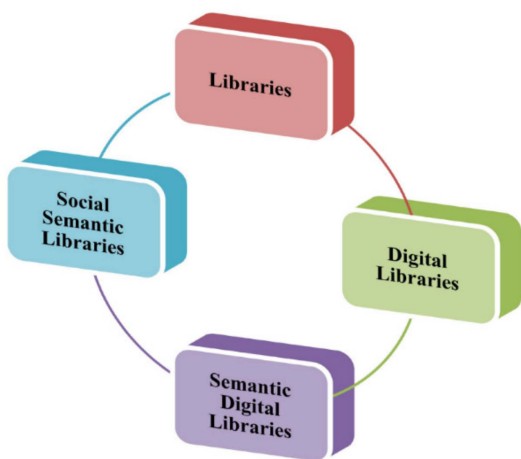

**Figure 5.** Various digital libraries utilized for semantic web applications.

The semantic web opens up the likelihood to adopt such a strategy. It offers open formats and regulations that can empower merchant unbiased arrangements, with valuable adaptability (permitting organized and semi-organized information, formal and casual depictions, and open and extensible engineering), and it assists with supporting decentralized arrangements where appropriate. Consequently, RDF can be utilized as a typical exchange position for index metadata and some shared controlled vocabularies, which can be utilized by all libraries and web search tools over the web. Table 4 shows the transition from conventional libraries to social semantic digital libraries.

**Table 4.** Transition from conventional libraries to social semantic digital libraries.

| References | Components | Stored Metadata | Web Interface |
|:---:|:---:|:---:|:---:|
| [44] | Library | Bibliographic cards | Note Books, Ledger |
| [35,45] | Digital Library | Database and digital bibliographic methods | Full-Text Search |
| [35] | Semantic Digital Library | Semantic based bibliographic methods | Searching, Retrieving and Storing through the ontologies |
| [46] | Social Semantic Digital Library | Bibliographic through appropriate annotation by end-users | Collaborative Search and Filtering |

### 2.3. Utilization of Metadata

Metadata are a key part of the arrangement of online lists that are accessible over the web. To utilize the semantic web to its best impact, metadata should be distributed in RDF

positions [29]. There are a few activities associated with characterizing metadata guidelines in the library and distributing network, including:

- The Dublin Core Metadata Initiative gives a standard arrangement of machine-understandable formats and rules for their utilization. This presently has entrenched RDF-controlled vocabularies.
- MARC: The notable MARC group from the Library of Congress has XML standardization and representations.
- ONIX: The ONIX for Books Product Information Message is the universal standard for speaking to and imparting book industry item data in the electronic structure XML representation.
- PRISM: The Publishing Requirements for Industry Standard Metadata detail characterizes XML metadata vocabularies for the magazine, news, index, book, and journal.

Such guidelines can be utilized over the web to give typical metadata details in XML or RDF that can be utilized to increase and offer library inventories on the web. PRISM and Dublin Core [32] are now usable in the semantic web. MARC and ONIX [47] still need some groundwork for complete utilization; however, they could be utilized as a source to advance the metadata given on the Web.

### 2.4. Role of Controlled Vocabulary

Controlled vocabularies [48] are the crucial factors in deciding the appropriate document classification by order; furthermore, they pave the way for effective searching of documents in the pool of documents. Controlled vocabularies, for example, taxonomies, classifications, and thesauri, are the key parts for implementing the above-defined task and are largely provided with some set of grounding rules to classify the document according to the subjects listed. The standardized tools and formats represented in the controlled vocabularies help to deliver such thesauri on the semantic web, which has been a significant activity of the SWAD-Europe venture [49]. This gives a lot of standard configurations and devices for depicting controlled vocabularies and characterizations called the Simple Knowledge Organization System (SKOS). It likewise gives some example thesauri that utilize these organizations, and some exhibit programming to permit individuals and projects to peruse and choose terms from a thesaurus over the web.

### 2.5. Some Other Projects

There are numerous tasks and activities that are giving access to libraries over the web, some of which are utilizing the semantic web legitimately, and others in the background. Some significant ones include: The Open Archive Initiative [50], which gives direct access to organized metadata through its metadata reaping convention; the Simile Project, which utilizes the semantic web to upgrade between operability among computerized resources, schemata/vocabularies/ontologies, metadata, and administrations; and DELOS, a European Network of Excellence on Digital Libraries whose site gives a lot more connections to digital library ventures [45].

## 3. Semantic Web in Virtual Communities

Recently, the growth of the semantic web project has been phenomenal, and in virtual networks, people can distribute data about themselves, their inclinations, and their work, and permit similar people to find and offer those data to construct a virtual network of individuals sharing thoughts. The friend-of-a-friend or FOAF [51] venture gives a basic language that permits individuals to distribute data about themselves, their work, and interests, alongside their contact subtleties (with due regard to security). This is valuable yet becomes intriguing when individuals can likewise distribute connections to others they know in the network. Moreover, FOAF gives a system of connections between individuals. You can follow the degree and extent of the virtual network of people, finding new potential contacts and contiguous networks of intrigue connections. Individuals are taking up this plan to fabricate instruments, for example, FOAFNaut, which permits the investigation of

the associations between networks [46]. Accordingly, we have a case of a system impact inside the semantic web when basic instruments and modest quantities of data consolidate to shape something of more prominent worth.

Different devices are intended to permit networks to impart data and insight. Web logs are settled outside the semantic web, permitting individuals to distribute onto the web and others to remark. By carrying web journals into the semantic web, with an explanation, they can be incorporated inside semantic web data gathering, blending, and looking, so they can partake in a progressively coordinated manner. A case of this is the work on semantic blogging from HP Labs in Bristol [52], where websites are explained with the goal that data on lists of sources and perusing records can be shared, looked at, and talked about [17]. Conceivably, this gives a priceless shared asset of commented on reference materials for a network, for example, a gathering of scientists or understudies. Essentially, devices, for example, Annotea, use the comment in RDF [53] to give remarks and explanations on website pages, with the goal that remarks gave by the network lead to the conversation [1]. So also [54], the web-based news-syndication framework RSS (for either Rich Syndication System, or RDF Syndication System), gives a component to distributing, sharing, joining, clarifying, and looking through news records and conversation bunches, and a few renditions of RSS use RDF, including RSS 1.0. For instance, RSS is utilized by the Nature Publishing Group to keep researchers and data analysts regarding the most recent news from their diaries, utilizing a mix of Dublin Core and Prism metadata.

Network entries give the main issues where virtual networks can convey and share data, find new contacts, and remark on one another's work. Semantic web innovation is being utilized to build such entrances to give a more extravagant way to deal with sorting out and looking through network gateways, normally based on the semantic portal innovations above. A model is CSAktiveSpace [55] from the University of Southampton, which assembles data on the dynamic scientists in computer science inside the UK, classifying their examination subjects and rating them on yield. This instrument gives a rich interface permitting the client to investigate the computer science people group inside the UK from various points, including some unordinary search interfaces permitting land look, for example, discovering specialists on neural networks working in Scotland.

Another way to deal with network entries is given by the Semantic Web Environmental Directory (SWED), also from HP Labs in Bristol [56]. This site unites data about natural associations in the UK, enormous and little, from the RSPB to neighborhood untamed life perception gatherings, which again can be looked at in an organized way, with the goal that clients can quickly distinguish the gatherings that most intently coordinate their prerequisites. A fascinating element of the plan of this framework is that as opposed to being overseen midway, every association is answerable for entering and keeping up its data in a dispersed manner, which is then totaled together. As every association has a personal stake in staying up with the latest, there is a more noteworthy possibility that the entryway will stay current with little exertion concerning the focal host.

FOAF has reached out to help community-based trust systems, particularly crafted by Jennifer Golbeck at the University of Maryland [33]. The thought here is to tell other people whom you know, besides the amount that you confide in them. By accumulating each other's assessments of the people in the system, the network can distinguish reliable people. This is much the same as the 'notoriety the executives' framework, which is given by online administrations, for example, the closeout house on eBay. Unmistakably, such a methodology is available to maltreatment of slander and profoundly emotional predisposition (individuals rating their ex-partner as amazingly dishonest is not exceptional!); however, taken care of attentively, such a framework could be incredibly helpful to recognize network experts on specific subjects.

A key element that these network-based devices share is that they are modest, straightforward, and open. The foundation to help a network can be immediately amassed with little cost and with a moderately limited quantity of specialized ability. By conglomerating

data from various sources, new and unexpected associations and data can develop. Some of the potential tasks that the semantic web technologies can perform are listed in Table 5.

**Table 5.** Some of the semantic web tools and projects.

| References | Potential Task on Semantic Web | Tools/Languages Used |
| --- | --- | --- |
| [26,28,53] | RDF Conversion | Drupal, MARC Edit, FOAFcalm, Sesame, Virtuso, PoolParty, JavaAnzo, MARcont, etc |
| [29,32] | Available Metadata Schemas | DOAP, EAD, MODS, FRBR etc. |
| [27,36] | Free Searching Tools | Swoogle, Twinkql, RelFinder, Lucene, CiteSeerx, GFacet, etc. |
| [57] | Information/Pattern Filtering Tools | Triplify, Virtuso, PoolParty |
| [35,39,58] | Ontology Development | Protégé, RDF2Go, SW MediaWiki |
| [37,45] | Some SW Portals/Projects | PubMed, BRICKS, NSDL, Google Books, SWAD, CACAO Projects, FEDORA, Whi, etc. |

*3.1. Semantic Web in Virtual Organizations*

An increasingly thorough methodology is being taken when individuals and associations wish to officially work together towards shared objectives over the PC framework. Instances of this in the higher education (HE) and further education (FE) people group would incorporate exploration ventures with accomplices spread across Europe, or in PC helped to realize, where the potential students and professors wish to share web-based instructing and learning assets and participate in a group activity such as student team projects.

At first, the virtual organizations [59] were used to delineate the computer-based environment, which can be shaped across authoritative limits. The well-formed standards and measures are rising inside the web and grid technologies to give the foundation to help virtual associations, ordinarily utilizing the idea of a service-oriented architecture where projects and emerging tools provide standard web-based access to support different individuals from the joint effort. To help these virtual organizations, there should be ways for individuals from the joint effort to discover the offered benefits generally proper to their necessities, arrange their utilization in the certainty that malignant demonstrations will not happen, and co-ordinate the utilization of different administrations to give the ideal outcome. These standards and measures require the advancement of some general vocabularies and arrangement conventions. The semantic web can give a fundamental structure to permit the arrangement of administration design to help virtual organizations. This idea is present in some cases given the depiction of the semantic grid [60]. Altogether, for a help to be utilized, it should be found. Likewise, the necessities of the client should be accommodated with the abilities of the administration. Certain revelations and discoveries could utilize semantic representations, utilizing an RDF explanation of the administration interfaces. These announcements ought to be machine-understandable capabilities, extensible depictions to help whether assistance can play out a given errand, and what sort of execution the administration can give. At that point, we could witness whether the enabled service discovery fulfills the client prerequisites. The DAML-S and the web service modelling ontology ventures [37] are investigating plausible ways to deal with the issue of giving semantic comment to service administrations.

A significant part of virtual organizations is the certainty that the pernicious utilization of service administrations is withheld, and again, the semantic web can bolster secure access to certain services. Some of the underlying endeavors in the utilization of semantic web formats for fundamental security applications (for instance, validation, grant permissions, user authentications, trust management, and key encryption) have started to tolerate the organic product. For instance, Denker et al. [61] have coordinated a lot of ontologies and security expansions for web service profiles. The authors [62] had additionally created Rei, a semantic-web-based approach language. Moreover, KAoS services consider

permitting certain access specifications, conflict resolutions, and policy management of organizational environments.

Eventually, virtual organizations are to permit clients to consolidate benefits together. For instance, a scientist or active researcher may wish to divert the potential data from service at a specified hotspot, to an investigation device at a subsequent area, with the consequences of the examination diverted to a data visualization tool at a destined location. These services should be coordinated. The appropriate workflow management has been developed over the most recent years as a method of planning business, and measures, for example, BPEL4WS is rising [63]; these should be alternate ways to permit the synthesis of service administrations. The Web Services community is starting to think about how to scale out the semantic representation to permit the amalgamations of Web Services.

The exploration, advancement, and establishment of grid foundation is continuing quickly in the UK under the National E-Science Program; thus, a system framework supporting virtual organizations is turning into a reality for some analyst's overall orders. There is expanding enthusiasm for investigating how it may be best misused to help to instruct, with tasks, for example, the European Learning Grid Infrastructure taking a lead [64]. Throughout the following few years, the model consequences of this work are probably going to show up in the up-and-coming age of virtual research environments.

### 3.2. Semantic Web in E-Learning

The semantic web has enormous applications to the e-learning platform, for both local and distance training. The thought of a 'learning object' as a distinguishable unit of online digital material that can be reused and joined with other learning objects has been a focal component of e-learning frameworks. This idea has been censured for being excessively unyielding and not considering the specific adapting needs of people or the prerequisites of setting and accentuation of instructors. Notwithstanding, utilized appropriately, it is a helpful and influential idea and one which the semantic web has a lot to offer. It has been largely stated that the learning objects can be composed of huge data hubs and shared across distributed (P2P) systems. The Edutella venture [65] is trying to give an RDF-based P2P construct for sharing learning objects. People can distribute learning items to the system (see Table 6), giving rich metadata that are 'enlightening data about learning assets for the reasons for discovering, overseeing, and utilizing these learning resources more adequately.

**Table 6.** Ontology for E-learning platforms.

| References | E-Learning Features | Semantic Web Options | Limitations |
|---|---|---|---|
| [24] | Semantic web languages and technologies | RDF, RDFS, OWL, SPARQL, XML, SWRL | Connects only with real-world entities. |
| [59] | E-learning standards and specification | IEEE LOM, IMSLD, IMSQTI | Deeply focused on fixed standards and relied heavily with open specifications. |
| [65] | Suitable virtual learning | Service-oriented architecture, web services, SCORM | Learning objects are light weighted and web services are domain specific. |
| [66] | Potential information in virtual learning environment (VLE) | Content, metadata, instructor, system, etc. | Data exchanges have been operated between trusted sources. |
| [45] | Interface for learners | LMS, web-based system, hypermedia system, etc. | Learners' authentication can be verified through web-based systems or any LMS only. |
| [57] | Application of ontology | Ontology deployment, adaptability, interoperability and appropriate annotation of data. | Ontology deployment is restricted for new frameworks and models. |
| [30] | Ontology inclusion for VLE | Online learners, instructor, pedagogical interface, domain specification, etc. | Interoperability between two ontological frameworks is restricted. |

At that point, the shared data repository can be looked at and articles can be recovered dependent on their semantic comments. Rich semantic representation dialects for learning objects are showing up. For instance, the Educational Modelling Language (EML) [67], the IMS Global Learning Consortium's proposed set of incorporated measures for e-learning subjects, including a metadata specification and the learning object metadata (LOM), a standard characterized by the Learning Technology Standardization Committee (LTSC) of IEEE [66]. All these areas of today are characterized in XML; however, they are versatile in RDF for use in the semantic web. This will permit a more extravagant collaboration with the learning material, with a semantic ontology-based argument for arranging the necessities of students to the accessible learning materials. Once more, we are probably going to see, in the following few years, the presentation of semantic web innovation into virtual learning environments, right off the bat at a test stage, and afterward more profoundly implanted. Illustrative setting and understanding into the advancement of information can be given by 'information diagrams', characterized by Stutt and Motta [66] as 'pathways through debates and accounts and different structures, for example, analogies and compositions of logical standards'. Once more, RDF and other semantic web advancements give the regular medium to speaking to and conveying such diagrams. Figure 6 shows the ontological framework for an E-learning system.

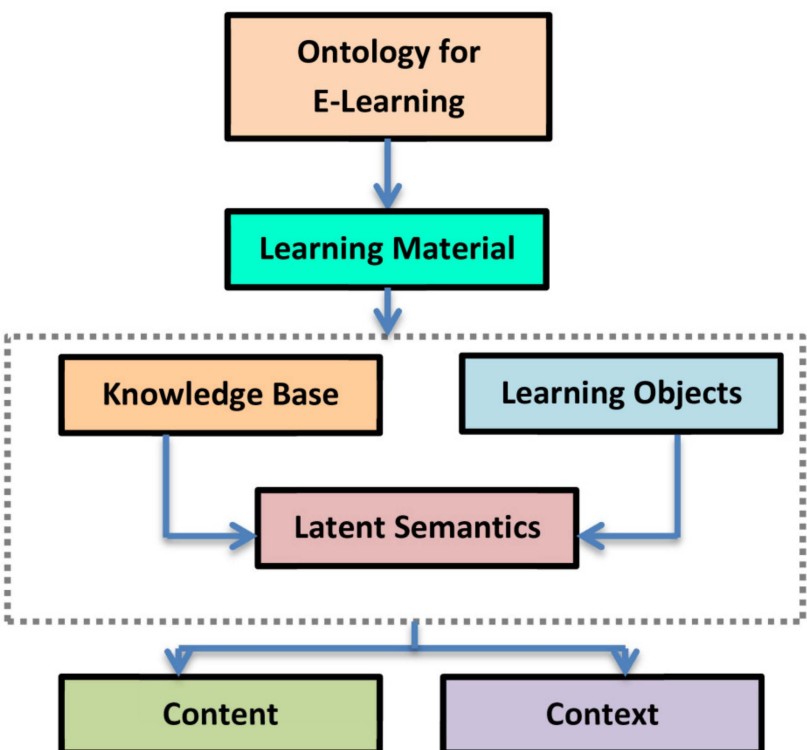

**Figure 6.** Ontological framework for E-learning system.

## 4. Ontology-Based Information Retrieval

This section surveys the best of ontology-based database data recovery. Here, an authentic outline of data recovery approaches is first introduced, trailed by a nitty-gritty investigation of prevailing ontology-based query frameworks and information search methodologies corresponding to three diverse key viewpoints that guided the survey of such work. The three perspectives are: (1) ontology-based effective query construction, (2) ontology-based Information processing, and (3) query refinement with domain ontology.

### 4.1. Information Retrieval from a Historical Perspective

Information retrieval is the quest for data in databases. The requirement for successful techniques to computerize data recovery has developed in significance on account of the

huge increment in the measure of both organized and unstructured data epitomized in data sources. Throughout the years, numerous visual IE [68,69] approaches appeared which intend to diminish the end client's exertion while associating with databases. These methodologies expect to remove data from databases utilizing visual devices. Such methodologies incorporate query by example (QBE) or query by template (QBT). These methodologies work for essential relational database systems, principally because the plain structure of the database fits well with the even skeletons utilized in query interfaces. In any case, such methodologies do not help in semantic information extraction, nor do they give any question plan backing to produce complex queries. To remove the inadequacies encountered above, a further extension was upheld and suggested. One such model is QUICK (Universal Interface with Conceptual Knowledge) [70], which centers on robotizing query plans by misusing the ER logical plan. However, in reality, the ER model has been utilized essentially for database structure and they regularly do not store information about a particular domain. The ER-based query optimizations [38] cannot give a solid strategy to rely upon its exhaustiveness in communicating low-level query requirements. On the contrary, many ontology languages with appropriately determined semantics have been created. Some of the ontological tools used to perform the semantic search operation are been given in Table 7.

**Table 7.** Standard ontological tools for semantic search.

| References | Ontology Tool | License | Language |
|:---:|:---:|:---:|:---:|
| [71] | Apache Fuseki | Apache License | Java |
| [72] | Protégé | W3C | Java |
| [73] | Virtuoso | GPL | C |
| [74] | Sesame | BSD | Java |
| [75] | Blazegraph | GPL | Java |

*4.2. Ontology-Based Query Formulation Approaches*

Ontology-based interactive frameworks are querying refinement systems for databases that utilize visual descriptions to communicate related data requests. These frameworks were able to adjust ontologies for database query plans to improve the viability of the human–PC interactions. As of late, a number of such frameworks have been accounted for in the literature [58] (e.g., TAMBIS, GRQL, SEWASIE, Ontogator, OntoViews, OntoQF, VISAGE, Smartch, Semantic-based and numerous others). In a large portion of these ontology-based interactive frameworks, the search queries are performed utilizing an ontological design that connects the ontology as a tree structure. The real pursuit is performed employing idea choice through a visual tree or watchwords commented on by the visual ontological concepts. The TAMBIS framework [57] bolsters the specialization or speculation of the base or filler ontological concepts to construct database-explicit questions intuitively. Here, the information in the databases is put away (connected) as occasions of ontological ideas. This methodology can be applied to determine combination issues, where all data sources have a similar mapping or give almost a similar perspective on an area.

Another comparative methodology dependent on ontological diagram design questions [44] is introduced in GRQL and KnowledgeSifter. GRQL depends on the full intensity of the RDF/S information model and gives a GUI to building inquiries dependent on the ontological route. In this methodology, questions are developed by graphically exploring through individual RDF/S classes and property definitions. In SEWASIE (SEmantic Webs and AgentS in Integrated Economies) [58], the standards of structuring and building up an efficient ontology-based query interface are introduced. The query interface of SEWASIE underpins the client in detailing a question through an iterative refinement process upheld by ontological concepts where in the query actual plan process, a client can determine a request using some conventional terms, can refine a few terms of a query or can present new terms, and can repeat the procedure if necessary. In OntoQF [58], OWL-DL ontologies

have been utilized for data retrieval via naturally creating social database queries utilizing off-the-shelf domain-specific information.

In contrast with other existing methodologies, one of the primary highlights of the OntoQF approach is that it utilizes a blend of both database-to-ontology substitution and mappings to empower the programmed query detailing process, which helps in creating exact database processes. Generally, OntoQF utilizes a two-stage approach. In the first pre-processing stage, the efficient domain-specific ontology is created from a social blueprint and related mappings are characterized, which connects the ontological concepts to social entities, and vice versa. Moreover, the ontology domain knowledge is to be communicated regarding OWL-DL declarations as concepts, which should be predictable with the particular area ontological outline. In the subsequent interpretation stage, the OntoQF process interprets ontological explanations into comparing social query refinement systems. The OntoQFs approach is reasonable for those frameworks or information mining applications that mean to keep all information at the first location(s) and use domain-specific knowledge for information-based data.

The framework introduced in [30] gives an intuitive database query refinement system through an undirected graph which encourages natural dialects. The ontology language utilized in this framework depends on the RDF structure. To build an effective query-based system, the given search terms are proposed to a client in a characteristic language from a predefined jargon. In a report of the EU Translational Research and Patient Safety in Europe (TRANSFoRm) [76], a question and information extraction workbench has been introduced. The TRANSFoRm question detailing workbench programming instrument gives interfaces to creator, store, and send inquiries of clinical information to recognize subjects for clinical investigations. Furthermore, TRANSFoRm inquiry plan workbench empowers clients to characterize rule bunches deftly while cooking for complex queries with blends of administrators.

*4.3. Ontology-Based Information Linking Approaches*

The work performed in the European TONES venture [30] gives relational database access through ontologies. In this methodology, effective information processing is empowered by characterizing joins between ontological terms and relational data. This ontology-to-database planning empowers an originator to connect an information source to an OWL-Lite. While characterizing mappings, the planner needs to consider that a specially appointed identifier ought to signify every concept and none of the ontology instances can be mistaken for data items in the preoccupied knowledge source. The queries are detailed by counseling ontology-to-database planning rules; however, this standard determination process is performed physically by ontology and database specialists. This methodology stores ideas from an information source as a major aspect of the ontology and connects real information with ontological ideas. The query refinement processes are improved by utilizing the semantic information system in a domain-specific ontology. Database query systems are changed by utilizing is-a relation, part-of, and having-of connections between existing ontological concepts.

The work completed in structuring ontology-based intelligent data recovery interfaces [36] gives a web data recovery framework. This methodology fills in as an intelligent data recovery framework where clients are guided through an OWL-based driven graphical interface to characterize the pursuit standards. This work addresses the issue of "where to begin in the use of an ontology-based IR interface"; that is, which components of the ontology ought to be given to the client to start the searching detail [44]. Suitably, a client initially chooses significant domain knowledge to begin fabricating an existing query. The interface at that point gives various query passage focuses alongside their representations. When the client chooses the domain-specific ontology components, web data components are recovered by following the static domain ontology-to-web connections.

In the SemanticLIFE venture [77], a front-end approach controls the clients in creating information demands. The SemanticLIFE framework incorporates numerous information

sources and stores them in an ontological archive. The virtual query segment of the SemanticLIFE framework permits semantic question composing on the ontological RDF-based store. Clients are given an outline about the framework information through a virtual data part that stores the removed metadata of the information sources as ontology. The methodology gives an existing query system, which prescribes the query designs as per the clients' questioning setting. Since it depends on a typical ontological design from the nearby information source ontologies, this methodology can refine clients' queries and make sub-queries over the information sources.

*4.4. Ontology-Based Query Refinement Strategies*

The ontology-based query refinement approaches were mostly targeted on empowering users to make an enhanced version of formulated query. These methodologies endeavor to further develop data recovery by supplementing or adding additional terms into an underlying question. The vast majority of the current query refinement approaches incorporate both question modifying and extension activities. Utilizing these methodologies, the users furnish collaboration with candidate terms that are dependent on concept hierarchies and the root of the terms normally from the created space ontologies and related ontological construction. This part examines these ontology-based query refinement procedures that have been presented in the course of recent years, for example, thesaurus ontology navigation, ambiguity driven, information need driven, and so forth. Majorly, these methodologies utilize the WordNet structures to fit the correct senses for the candidate terms that have been selected for query refinement process.

One more methodology dependent on this ontology framework is the knowledge sifter [44]. The knowledge is an adaptable specialist-based framework that supports admittance to heterogeneous data sources and depends on the specialists' innovation for query refinement. In this methodology, a user query is normally detailing specialist upholds user query to get to various ontologies utilizing a coordinated applied model communicated in the OWL. This user query plan specialist additionally counsels the ontological framework to refine or to sum up a query dependent on the semantics given by the ontology.

In QuOnto and MASTRO [78], the query responding to the process is performed through query modifying. In these methodologies, the user queries are first reformulated based on ontological intentional information, and afterward, they are assessed by a data set motor utilizing a method for predefined mappings. Information base perspectives are characterized for ontological ideas, and jobs utilizing SQL queries are determined in thesaurus-to-dataset planning revelations.

In ontology-based device, a change over to a characteristic language query into nRQL query has been proposed. To accomplish the change, initially, a pre-populated word reference is utilized to look through the equivalents of query terms. Assuming no coordinating with records is found, the ontology search is performed, which brings about removing a grouping of entities addressed in type of triples. At long last, nRQL query is produced dependent on the resultant data. The ontology-based query refinement approaches, for example, step-by-step query refinement, analyze query equivocalness corresponding to both primary and semantic ambiguities. Primary uncertainty manages the genuine construction of a query that is dissected as for the fundamental ontological information. For the situation where a contention is distinguished, elective ideas are recovered and introduced to the user for determination.

## 5. Semantic Web in Healthcare

The amount of data generated in biomedical research has been progressing rapidly over the years recently, and the inherent capabilities of the digitization of this booming medical sector is also gaining huge momentum after the severe impact of COVID-19. The healthcare data have been produced largely from diverse sources such as hospitals, research labs, medical images, clinical records, and some patient-monitoring systems. Although

the data are growing exponentially at one end [17], it has not been a bigger challenge for technologies to gather, store, and analyze the medical records and further made simpler to infer some meaning insights from the medical data. Moreover, the technologies have paved way for transparent storage of medical data and affordable retrieval of medical results. However, there are some concerns related to utilizing this healthcare data for further processing in some distributed environments, and the prominent issues addressed in healthcare data are mostly in interoperability aspects. Actually, the clinical information has been stored in multiple locations with different managerial divisions such as emergency clinics, various states, stakeholders, etc. The unification of the healthcare data is a cumbersome process and the reconciliation on the data is a challenge process to any existing technologies. As the data have been gathered from different ends and verticals of the healthcare domains, the difficulties in accessing the data would be majorly on its naming conventions, inherent structures of the medical data [18], the basic organization of the report, and finally, the format in which it has been properly rendered for unification. Hence, to meet the above challenges faced in the healthcare domain, significant progress has been made in this field to augment the capabilities of integrating the data and make the data thoroughly consistent in all sorts of configurations. The role of semantic web in healthcare is gaining huge research attention; furthermore, it would always pave the way for easy searching operations, reusing the existing ontologies, integrating the resources for easy configuration, and ultimately sharing the information in the format that has been required for the applications.

Some of the important techniques exhibited by the semantic web technologies to meet the challenges faced in the healthcare domain were mostly efficient [19] and oriented towards comprehending the data formats fixed on different applications. Furthermore, the semantic web solutions have enabled the healthcare community to strenuously process the following strategies and streamline the issues faced earlier by the web-based technologies.

(a)   Facilitate the healthcare domain experts to understand the internal process of the semantic-web-enabled applications and use the right signs and symptoms of the disease.

(b)   Provide the appropriate learning to the patients to avail the application rendered for their use during their clinical investigations and provide the reports required for the further examinations and cross-validations.

(c)   Establish the proper setup between the doctors and patients to deliver the right clinical investigations and provide the necessary reports to the patients wherever deemed necessary.

Some of the fundamental techniques used by semantic web technologies for utilizing the healthcare data are ontologies. To support the ontologies for the healthcare domains, the following steps have been taken for efficient search and integrations.

(a)   Convert the healthcare data to appropriate RDFs by identifying the potential medical entities defined by the ontology.

(b)   The suitable RDF schema has been laid out to set the proper business rules and include the hierarchy that is made perfect for integrations. Furthermore, it enables to transform the original healthcare into unified RDF representations.

(c)   Eventually, store the processed data into the system using OWL.

(d)   The integrated RDF data can be retrieved using SPARQL query and obtain the information semantically.

This process has enabled the data to transform it to a new format that is RDF format and thereby increases the chance of data availability and also provides the users to effectively choose the data that they were requested for utilization and discover the inherent details that have not been investigated using the earlier web-based applications [20]. The semantically processed healthcare data can also be used for data visualization, and it helps to discover some hidden patterns lurking in the medical reports submitted to the application. This semantic integration helps to navigate further into the system to capture

the models and querying the data that has been mapped into some named RDF graphs to obtain different degrees of data granularity.

Some of the standard ontologies used in the healthcare domain are listed in Table 8.

**Table 8.** Standard ontologies used in the healthcare domain.

| Reference | Healthcare Ontology | Details | Format | Total Classes |
|---|---|---|---|---|
| [32] | ATC | This ontology can gather details related to drug ingredients of the organ and classify it using the chemical characteristics. | UMLS | 6.358 |
| [62] | DOID | General human illness related ontology and segregate the details of the illness on some medical properties. | OBO | 12,694 |
| [67] | HP | Classify the monogenic diseases and set the medical vocabulary straight for phonetic highlights. | OBO | 18,407 |
| [66] | MedDRA | Drug classification and discovery of health consequences. | UMLS | 73,429 |
| [44] | PMR | Rehabilitation data for patients. | OWL | 1597 |

## 6. Open Challenges

Although the data have been gathered from diverse sources and different platforms, there needs to be manual interventions to interpret the data to make them meaningful. Furthermore, there is a lack of a unified approach to integrate the common data and filter the disambiguates persists on the captured datasets. The semantic web applications have been facing some serious challenges to convert the unstructured data or semi-structured data into a standard open format. The need for an open knowledge graph is highly recommended in this regard and we now solely depend on it with large knowledge bases such as DBpedia or YAGO for effective disambiguation. A unified model is required to answer the complex queries raised by the users and optimize the processes to promptly reply to the questions. In order to achieve this task, it has faced the following open challenges tabulated in Table 9. Figure 7 portrays the open challenges for using the semantic web.

The quality of data has not been a big problem for dealing with projects related to the semantic web and converting the unstructured or semi-structured data format into fully structured format is made affordable using RDF schema and OWL frameworks. Hence, most of the researchers have not faced any issues with data formats, and they have extensively used the data formats retrieved from some of social media sites. However, we observed that resource consumption and resource sharing have been facing some serious issues due to the fact that there is no unified approach to share the data to diverse web applications and the resource consumption is very poor because of high latency between two or more heterogeneous web systems. Data integration and reusability of data have nearly made a huge bottleneck for data sharing and pose great difficulty in accessing information from diverse sources. In the ontological framework, the words may climb into different concepts of the hierarchy. For instance, the term "Apple" may represent the fruit and follow the hierarchy related to "Fruits" and it may also refer to the mobile brand "company" and follow the concepts of "Electronics". This sort of ambiguity is becoming prevalent in resource sharing and poses great challenges for the real-world web application to retrieve the correct information for the user request.

**Table 9.** Open challenges for using semantic web (✔: available, X: not available).

| References | Quality of Data | Resource Consumption | Service Interoperability | Data Integration | Resource Sharing | Reusability |
|---|---|---|---|---|---|---|
| [24,26] | ✔ | X | X | ✔ | X | ✔ |
| [42,49] | X | ✔ | ✔ | ✔ | ✔ | X |
| [51,52,61] | ✔ | X | ✔ | X | X | ✔ |
| [64,68,69] | ✔ | X | X | X | ✔ | X |
| [57,77] | ✔ | ✔ | X | ✔ | X | X |

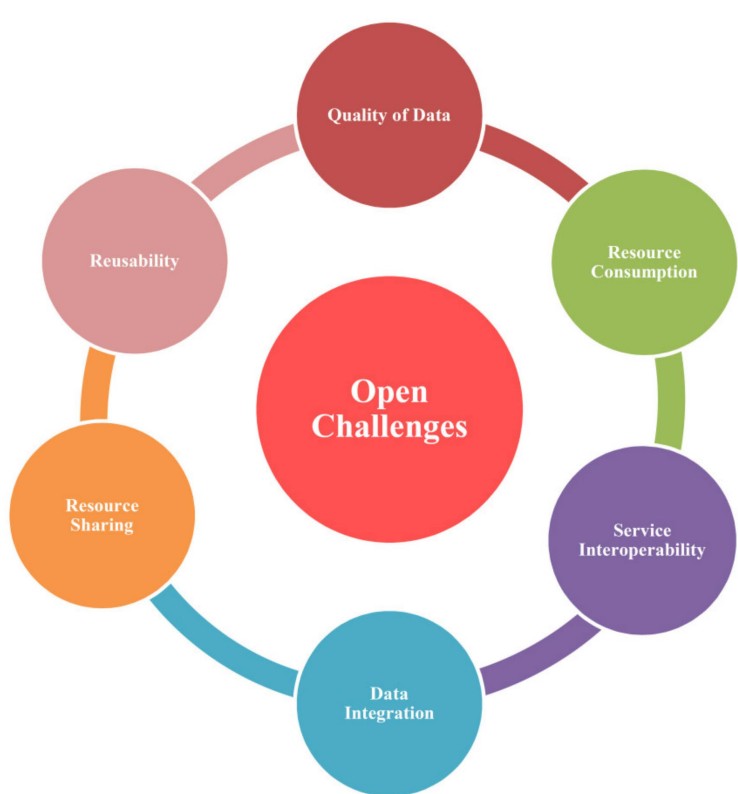

**Figure 7.** Open challenges for using semantic web.

## 7. Road Ahead for Semantic Web/Future Directions

It has become very evident that the objective of the semantic web has reached its core potential in terms of offering a wide range of solutions such as data sharing, knowledge discovery, service integration, and ontology reusability. These solutions provide the tasks to complete it at ease and achieve with the task of knowledge extraction with good precision. It has been demonstrated in the knowledge extraction processes such as knowledge graphs, schema.org, and some of the life-science ontologies. Still, the prominence of the semantic web has not reached its peak and further requires more advancement in the subfields of the semantic web. As the wealth of knowledge grows exponentially at one end, there are certain lapses in efficient data management and a lack of standard prototypes to fit into the well-designed applications. The emerging practitioners of the semantic web have mostly been confronted with diverse problem-solving approaches and also struggled with a cacophony of sub-problem integrations. Hence, the semantic web requires a huge consolidation to be able to infuse the application-driven orientation into these sub-fields. The application-oriented processes should be well-documented so as to make their goals precise, and made affordable for easy-to-use, enabling the well-integrated tools to supplement the entire requirements of the system. For instance, when it takes for effective consolidation of data and put the hierarchy of data integration, we need to go for popular ontology software such as Protégé Ontology Editor, OWL API, ELK reason, which is easy to work and offers the

reliable serialization of RDF and OWL. Some of the real-world applications of the semantic web have been depicted in Figure 8. The domain-specific applications of the semantic web have been becoming wider in recent years after the infusion of machine learning models and deep neural networks. The influence of utilizing the semantic web frameworks has even dominated in fields such as astrophysics, drug discovery, biological sciences, etc.

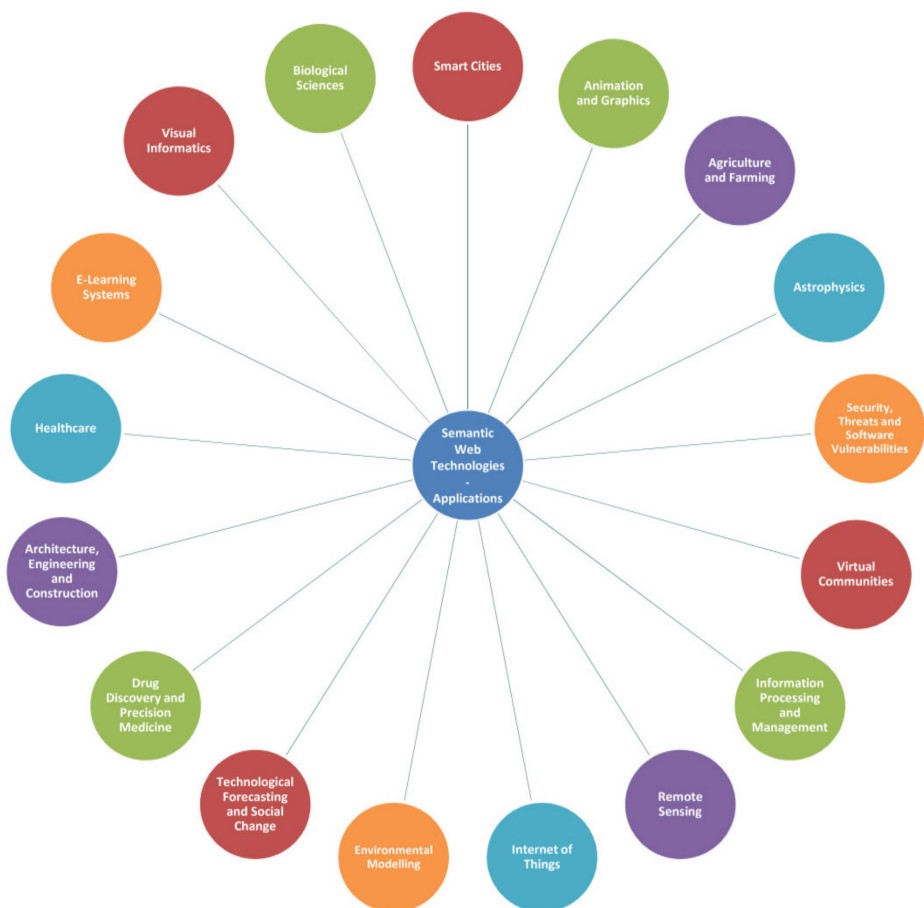

**Figure 8.** Real-world applications of semantic web technologies.

For the future directions, semantic AI is gaining huge popularity in recent years and an enormous amount of research has been happening in this field. Semantic AI is said to be the next-generation artificial intelligence for many real-world applications. Semantic AI can deploy the knowledge graph effectively and improve the searching process at the speed of convolutional neural network. Semantic AI has the inherent capabilities to pass through the corpus-based ontology learning and internalize the mapping process based on the edges created in the knowledge graph. This integrated approach followed in semantic AI can lead the system to be transparent and provide the cutting-edge solutions for the underlying knowledge models. It has been widely believed that the integration of semantic AI into any established organization would sustain the business to a lucrative framework and will lead to new model of AI governance.

Some of the future directions and research areas relied on with semantic web are listed below in Figure 9.

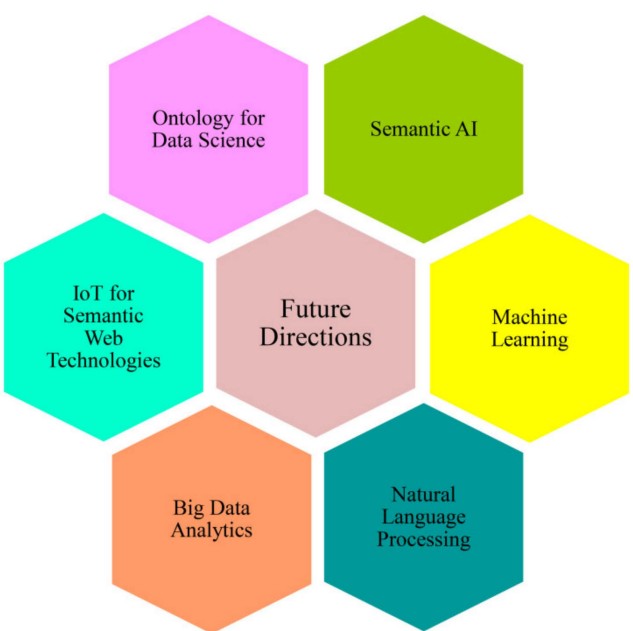

**Figure 9.** Future directions for semantic web technologies.

## 8. Conclusions

Over the 20 years of semantic web existence, its importance has been widely recognized in wide spectrum of knowledge management and, particularly, it has taken the strong base in data sharing, knowledge discovery, integration, and reusability. The contribution of semantic web has been growing considerably using the deep-rooted ontologies and other ontology-modelled applications [79–126]. In this research survey, we focused more on the utilization of semantic web technologies in healthcare, virtual communities, and how the information retrieval task has been performed effectively to find the hidden knowledge and establishes the strong contribution by utilizing ontological sources such as Wikidata, DBpedia, and Schema.org. The emergence of semantic web technologies has provided the hassle-free application interface to reduce the hidden relationships among the potential real-world entities and increases the interoperability among the concepts. It has been argued that the integration of the semantic web and artificial intelligence would pave way for a unified approach to deal the disambiguation problems faced in unstructured data.

The semantic web technologies have given many opportunities to transform the unstructured or semi-structured data formats into some standard structured format using the RDF schema and OWL frameworks. Furthermore, the semantic web and knowledge management can complement each other for resolving the ambiguities persisting in the text documents and addressing the challenges with a high precision rate. In this survey paper, we have highlighted the issues and problems faced in retrieving the potential information from diverse sources such as healthcare, virtual communities, and other textual data formats. We have opened up some of the serious research challenges and explored the potential solutions for the problems related to healthcare and other information-processing systems.

**Author Contributions:** Conceptualization, S.K.N.; methodology, K.S.; software, S.K.N. and K.S.; validation, Y.-C.H. and K.-Y.H.; formal analysis, S.K.N.; investigation, S.K.N. and K.S.; resources, Y.-C.H. and K.-Y.H.; data curation, S.K.N. and S.K.M.; writing—original draft preparation, S.K.N. and K.S.; writing—review and editing, S.K.N., K.S., Y.-C.H., S.K.M. and K.-Y.H.; visualization, K.S.; supervision, K.-Y.H.; project administration, Y.-C.H.; funding acquisition, K.-Y.H. All authors have read and agreed to the published version of the manuscript.

**Funding:** The authors thank the Ministry of Science and Technology, Taiwan, for financially supporting this research under Contract MOST 110-2221-E-005-067-.

**Conflicts of Interest:** The authors declare no conflict of interest.

## Appendix A

List of abbreviations used in this manuscript along with their full form.

| Acronym | Definition |
| --- | --- |
| EML | Educational Modelling Language |
| FOAF | Friend Of A Friend |
| LOM | Learning Object Metadata |
| LTSC | Learning Technology Standardization Committee |
| NLP | Natural Language Processing |
| OWL | Web Ontology Language |
| POS | Part of Speech |
| QBE | Query By Example |
| QBT | Query By Template |
| RDF | Resource Description Framework |
| RSS | Rich Syndication System |
| SKOS | Simple Knowledge Organization System |
| SWED | Semantic Web Environmental Directory |
| URI | Uniform Resource Identifier |
| HE/FE | Higher Education/Further Education |

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
