# Peer review of "A Contemporary Review on Utilizing Semantic Web Technologies in Healthcare, Virtual Communities, and Ontology-Based Information Processing Systems"

_electronics, doi:10.3390/electronics11030453_

Round 1

Reviewer 1 Report

In this paper the authors review about semantic web technologies where is applied AI in health care. The paper is structure in 7 chapters, starting with a detailed introduction and followed also with an extensive two chapter about semantic web technologies and semantic web in virtual communities which make up two-thirds of the work. About semantic Web in Healthcare is passing quickly in chapter 5 and about AI it is mentioned only twice in the paper (21 and 884). Also, the conclusions are extremely brief.

In my opinion, this paper describes the semantic web technologies in general sense and is mentioned and its use in healthcare is mentioned. I suggest to remove the AI from the title because it is only mentioned about it in this area.

I also suggest a detailed description of the scientific elements brought to us in the context of the title of the paper.

Author Response

*Reviewer 1 *

Comment 1: In this paper the authors review about semantic web technologies where is applied AI in health care. The paper is structure in 7 chapters, starting with a detailed introduction and followed also with an extensive two chapter about semantic web technologies and semantic web in virtual communities which make up two-thirds of the work. About semantic Web in Healthcare is passing quickly in chapter 5 and about AI it is mentioned only twice in the paper (21 and 884). Also, the conclusions are extremely brief.

Response: Thank you so much for the positive feedback. We have improved the manuscript based on the reviewer’s comments and suggestions. The deployment of semantic web technologies in various domains has been doomed diverse and made huge impact in many real-world applications. The Semantic Web is also a subsection of Artificial Intelligence and the purpose is to make the machine to comprehend the human language at ease. Hence, we had not focused much on Artificial Intelligence and restricted with the core applications of Semantic Web. But your point is valid and we oblige to your suggestions. Hence, we have decided to remove those sentences delineating about the role of Artificial Intelligence in Semantic Web. Also, we have updated the conclusion with more insights. Thank you for your suggestion.

Comment 2: In my opinion, this paper describes the semantic web technologies in general sense and is mentioned and its use in healthcare is mentioned. I suggest to remove the AI from the title because it is only mentioned about it in this area.

Response: Thank you for this valuable suggestion. As per the valuable suggestion, we have decided to remove the word AI from the Title and refined the new title as: A Contemporary Review on utilizing Semantic Web Technologies in Healthcare, Virtual Communities and Ontology Based Information Processing Systems”. 

Comment 3: I also suggest a detailed description of the scientific elements brought to us in the context of the title of the paper.

Response: Thank you for this valuable suggestion. Since the title has been refined into “A Contemporary Review on utilizing Semantic Web Technologies in Healthcare, Virtual Communities and Ontology Based Information Processing Systems“, we believe that it is not essential to include any scientific elements into the paper.

Reviewer 2 Report

This is a review paper describing the AI-enabled semantic web technologies in three fields. The paper is overall well organized and the English language is mostly fine.

For the content of this paper, I have several questions listed below:

1. AI technologies have gained rapid development in recent years and enhance many traditional techniques and fields. The title of this paper also emphasizes that the semantic web is AI-enabled. However, it seems that the authors only talked about AI part in Section 7 with only one paragraph. Explicit descriptions of how AI aided semantic web are suggested to appear throughout the text.

2. Table 2 made a comparison among existing surveys. However, the submitted paper itself is not included. Also, the "Main Focus" column is not intuitionistic enough. I would suggest refining it, i.e. splitting it to several more detailed columns.

3. For Figure 3 and Section 3-5,  the principles for grouping Section 3 & 4 together and separate them from Section 5 is not clear. Also, why choosing health care, virtual communities and ontology based information processing to survey, rather than other aspects of semantic web is not clear.

4. For Section 6, more descriptions on why the authors in Table 10 can not achieve all the 6 aspects are needed, rather than only one paragraph.

There also exist some mistakes in the text, e.g.

Line 104 score->scope

Line 176 a blank should be inserted between Section 1 and discusses

Line 176 discussess->discusses

Line 512 PC and Line 513 HE and FE are not explained in Table 1

Author Response

*Reviewer 2 * 

General Comment: This is a review paper describing the AI-enabled semantic web technologies in three fields. The paper is overall well organized and the English language is mostly fine.

Response: Thank you so much for the positive appreciation.

Comment 1: AI technologies have gained rapid development in recent years and enhance many traditional techniques and fields. The title of this paper also emphasizes that the semantic web is AI-enabled. However, it seems that the authors only talked about AI part in Section 7 with only one paragraph. Explicit descriptions of how AI aided semantic web are suggested to appear throughout the text.

Response: Thank you for this valuable suggestion. Since Semantic Web is also a subpart of Artificial Intelligence, we have not focussed much on the role of AI in Semantic Web Application. Based on your valuable suggestions, we have updated the title as “A Contemporary Review on utilizing Semantic Web Technologies in Healthcare, Virtual Communities and Ontology Based Information Processing Systems“.

Comment 2:  Table 2 made a comparison among existing surveys. However, the submitted paper itself is not included. Also, the "Main Focus" column is not intuitionistic enough. I would suggest refining it, i.e. splitting it to several more detailed columns.

Response: Thank you for this valuable suggestion. We have modified the “main focus” column as “One Phrase Summary” and we have also included a new column “Discussion on real-time availability of data” in the revised manuscript. 

Comment 3:  For Figure 3 and Section 3-5,  the principles for grouping Section 3 & 4 together and separate them from Section 5 is not clear. Also, why choosing health care, virtual communities and ontology based information processing to survey, rather than other aspects of semantic web is not clear.

Response: Thank you so much for this valid observation. Since the utilization of Semantic Web in Virtual Communities and ontology based information processing has been gaining huge momentum in recent years particularly after the devastating impact of COVID-19, both Virtual Communities and ontology based information processing systems are delivering the much needed services in this turbulent time. Hence, we have clubbed both in section 3 and 4. The Semantic Web in Healthcare is a different domain altogether and needs precise steps to explain how the Healthcare processing really works. Therefore, we have made it independently in section 5. We also listed some of the potential ontologies used for Healthcare applications.

Comment 4: For Section 6, more descriptions on why the authors in Table 10 can not achieve all the 6 aspects are needed, rather than only one paragraph.

Response: Thank you for this valuable suggestion. We have substantiated with more details of why the challenges has not widely been focused by some of the researchers. In Page 25, we have incorporated such changes in the paper. Thank you once again.

Comment 5:  There also exist some mistakes in the text, e.g.

Line 104 score->scope

Response: Thank you for this valuable suggestion. It has been updated. It’s a typo mistake. We apologize for the typo error.

Comment 6:  Line 176 a blank should be inserted between Section 1 and discusses

Response: Thank you for this valuable suggestion. It has been updated. It’s a typo mistake. We apologize for the typo error.

Comment 7: Line 176 discussess->discusses

Response: Thank you for this valuable suggestion. It has been updated. It’s a typo mistake. We apologize for the typo error.

Comment 8: Line 512 PC and Line 513 HE and FE are not explained in Table 1

Response: Thank you for this valuable suggestion. The expansion of HE and FE has been included in Table 1.

We once again would like to thank the reviewers for their constructive comments that helped to improve the quality of our work. We hope that our response is acceptable for the queries raised by the reviewers.

Thanking you,

Sincerely,

Authors  

Reviewer 3 Report

  • abbreviation listed in table 1 can be presented in a section with this name and not in the form of table 1
  • Figure 1 lacks good functionality
  • it is not clear why/how authors reached to 65 papers from 219 full text papers
  • Figure 4 requires a detailed caption to explain it
  • it is not clear how Figure 5 is illustrating the concept of digital libraries. both fig and its caption requires modification
  • A figure/flowchart listing all applications for semantic web is useful 
  • section 5 requires more relevant references 
  • it is not clear from its caption and details, that how table 10 is showing the challenges of semantic web
  • I recommend that authors better explain their contribution and highlight the need for such a survey paper
  • title is very long. a concise title can be more interesting  

Author Response

*Reviewer 3 *

Comment 1: Abbreviation listed in table 1 can be presented in a section with this name and not in the form of table 1

Response: Thank you for this valuable suggestion. As per the valuable suggestion, table 1 has been moved to Appendix A in the revised manuscript.

Comment 2: Figure 1 lacks good functionality

Response: Thank you for this valuable suggestion. As per the valuable suggestion, figure 1 has been modified in the revised manuscript.

Comment 3: it is not clear why/how authors reached to 65 papers from 219 full text papers

Response: Thank you for this valuable suggestion. We had reached to this final state of using 65 papers after gone through all the 243 research articles related to Semantic Web and filtered the research articles which are providing strong empirical evidences and the techniques and datasets used are completely different. We have not considered the papers which shares the same techniques and logics. This detail has been mentioned in the section 1.2.4 as well. Thank you very much for this keen observation.

Comment 4: Figure 4 requires a detailed caption to explain it

Response: Thank you for this valuable suggestion. We have now rephrased it in the revised manuscript.  

Comment 5: it is not clear how Figure 5 is illustrating the concept of digital libraries. both fig and its caption requires modification

Response: Thank you for this valuable suggestion. As per the valuable suggestion, Figure 5 caption is now updated. The figure talks about the inclusion of different digital libraries used to connect the potential named entities and find the context of the words using the metadata as well as through the shared indices.

Comment 6: A figure/flowchart listing all applications for semantic web is useful

Response: Thank you for this valuable suggestion. We have included a new figure (i.e. Figure 8) in Page#27.

Comment 7: Section 5 requires more relevant references

Response: Thank you for this valuable suggestion. As per the valuable suggestion We have added three relevant references into the section 5. Also, new references have been included in the revised manuscript.

Comment 8: It is not clear from its caption and details, that how table 10 is showing the challenges of semantic web

Response: Thank you for this valuable suggestion. Now we have added more content to substantiate the reason how it has become more challenging for the researchers when dealing the projects related to Semantic Web. In the page# 25, last paragraph, we have added the content to give the readers some clarity.

Comment 9: I recommend that authors better explain their contribution and highlight the need for such a survey paper

Response: Thank you for this valuable suggestion. As per the valuable suggestions, key contributions of this work have been included in the introduction section.  Also, in the conclusion, a new paragraph has been included, which explains the purpose of this survey paper and projects the impeding details related to use of Semantic Web technologies.

Comment 10: Title is very long. a concise title can be more interesting 

Response: Thank you for this valuable suggestion. As per the valuable suggestion, we have decided to remove the word AI from the Title and refined the new title as: A Contemporary Review on utilizing Semantic Web Technologies in Healthcare, Virtual Communities and Ontology Based Information Processing Systems”. 

We once again would like to thank the reviewers for their constructive comments that helped to improve the quality of our work. We hope that our response is acceptable for the queries raised by the reviewers.

Thanking you,

Sincerely,

Authors   

Round 2

Reviewer 1 Report

The authors took into account the comments made and made changes in this regard.

Reviewer 2 Report

The authors have made great efforts on modifying the paper according to the comments.